# Decoration of Anti-CD38 on Nanoparticles Carrying a STAT3 Inhibitor Can Improve the Therapeutic Efficacy Against Myeloma

**DOI:** 10.3390/cancers11020248

**Published:** 2019-02-20

**Authors:** Yung-Hsing Huang, Mohammad Reza Vakili, Ommoleila Molavi, Yuen Morrissey, Chengsheng Wu, Igor Paiva, Amir Hasan Soleimani, Forugh Sanaee, Afsaneh Lavasanifar, Raymond Lai

**Affiliations:** 1Department of Laboratory Medicine and Pathology, University of Alberta, Edmonton, AB T6G 2R3, Canada; yunghsin@ualberta.ca (Y.-H.H.); omolavi@ualberta.ca (O.M.); yuenw@ualberta.ca (Y.M.); chengshe@ualberta.ca (C.W.); 2Faculty of Pharmacy and Pharmaceutical Sciences, University of Alberta, Edmonton, AB T6G 2H7, Canada; vakili@ualberta.ca (M.R.V.); paiva@ualberta.ca (I.P.); asoleima@ualberta.ca (A.H.S.); fsanaee@ualberta.ca (F.S.); afsaneh@ualberta.ca (A.L.); 3Faculty of Pharmacy, Tabriz University of Medical Science, Tabriz P.O. Box 51664-14766, East Azerbaijan Province, Iran

**Keywords:** multiple myeloma, STAT3, S3I-1757, nanoparticle, CD38

## Abstract

STAT3 is an oncoprotein which has been shown to contribute to drug resistance in multiple myeloma (MM). Nonetheless, the clinical utility of STAT3 inhibitors in treating MM has been limited, partly related to some of their pharmacologic properties. To overcome these challenges, our group had previously packaged STAT3 inhibitors using a novel formulation of nanoparticles (NP) and found encouraging results. In this study, we aimed to further improve the pharmacologic properties of these NP by decorating them with monoclonal anti-CD38 antibodies. NP loaded with S3I-1757 (a STAT3 inhibitor), labeled as S3I-NP, were generated. S3I-NP decorated with anti-CD38 (labeled as CD38-S3I-NP) were found to have a similar nanoparticular size, drug encapsulation, and loading as S3I-NP. The release of S3I-1757 at 24 h was also similar between the two formulations. Using Cy5.5 labeling of the NP, we found that the decoration of anti-CD38 on these NP significantly increased the cellular uptake by two MM cell lines (*p* < 0.001). Accordingly, CD38-S3I-NP showed a significantly lower inhibitory concentration at 50% (IC50) compared to S3I-NP in two IL6-stimulated MM cell lines (*p* < 0.001). In a xenograft mouse model, CD38-S3I-NP significantly reduced the tumor size by 4-fold compared to S3I-NP on day 12 after drug administration (*p* = 0.006). The efficacy of CD38-S3I-NP in suppressing STAT3 phosphorylation in the xenografts was confirmed by using immunocytochemistry and Western blot analysis. In conclusion, our study suggests that the decoration of anti-CD38 on NP loaded with STAT3 inhibitors can further improve their therapeutic effects against MM.

## 1. Introduction

Recent studies have shown that nanoparticles (NP) can be an effective drug delivery system to treat cancers [1]. In addition to its usefulness in delivering hydrophobic drugs, NP can promote drug accumulation at tumor sites due to the fact that NP are too large to pass through the normal capillaries but small enough to leak through the poorly-formed vasculatures frequently present in malignant tumors [2]. To further increase the targeting ability and reduce drug toxicity, researchers have conjugated NP with various tumor-specific antibodies [3]. For example, different forms of NP conjugated with anti-human epidermal receptor-2 (HER2) antibodies have been generated to treat HER2-positive breast cancer [4,5,6,7]. Based on the results of a few studies, it appears that the decoration of NP with tumor-specific antibodies can indeed result in superior cellular uptake/cytotoxicity in vitro, as well as significantly improved tumor suppression in vivo, compared to their unconjugated counterparts [7,8,9,10,11,12]. Some studies also suggest that the conjugation of antibodies on the surface of NP is useful in overcoming drug resistance in cancers which overexpress drug efflux pumps (e.g., p-glycoprotein) [8,13]. The conjugation of antibodies on NP is versatile, and a variety of antibodies have been used to achieve specific experimental objectives, such as the use of antibodies targeting the vascular endothelial growth factor receptor (i.e., to block tumor angiogenesis) [12], matrix metalloproteinases (i.e., to block tumor invasion) [14], and transferrin receptor (i.e., to facilitate the crossing of NP through the blood-brain barrier) [15].

Multiple myeloma (MM) is a hematological disease which is characterized by a high frequency of relapses and resistance to chemotherapy. A signal transducer and activator of transcription 3 (STAT3), which is found to be active in more than 50% of MM, has been shown to contribute to the resistance to bortezomib, thalidomide, and dexamethasone in MM [16,17,18,19]. In view of its significance in cancer biology, STAT3 has been postulated to be an attractive anti-cancer target, and many STAT3 inhibitors (such as Stattic, S3I-201, and S3I-1757) have been developed [20,21,22]. However, the clinical utility of these compounds has been limited, which is likely related to some of their pharmacologic properties, such as their small size and hydrophobicity. Consequently, STAT3 inhibitors have been found to have relatively low therapeutic efficacy and high toxicity. In this regard, a clinical trial of OPB-31121 (an orally administered STAT3 inhibitor) in a cohort of patients with various types of solid cancer has reported that >80% of patients experienced significant nausea/diarrhea without therapeutic benefits [23]. To overcome these challenges, our group has recently generated NP to package STAT3 inhibitors. Specifically, we synthesized NP based on the poly(ethylene oxide)-*block*-poly(α-benzyl carboxylate-ε-caprolactone) (PEO-*b*-PBCL) backbone, and we used these NP to package S3I-1757 (denoted as S3I-NP); we found that S3I-NP exhibited significantly better anti-tumor effects in mice xenografted with a human melanoma cell line compared to free drugs [24].

In this study, we aimed to further improve the pharmacologic properties of S3I-NP by conjugating monoclonal antibodies against human CD38, a cell-surface marker highly expressed on MM cells, on the surface of S3I-NP (denoted as CD38-S3I-NP). We hypothesized that, due to the active targeting properties of anti-CD38 for MM cells, CD38-S3I-NP will demonstrate improved cellular uptake in vitro cytotoxicity and in vivo therapeutic efficacy compared to S3I-NP. Our results have provided the proof-of-principle that anti-CD38-conjugated NP loaded with STAT3 inhibitors are useful therapeutic agents for MM patients.

## 2. Results

### 2.1. Synthesis and Characterization of CD38-S3I-NP

To generate CD38-S3I-NP, we conjugated anti-CD38 monoclonal antibodies to the hydrophilic portion of PEO-*b*-PBCL. As illustrated in Figure 1A, anti-CD38 was first thiolated at the lysine residue located in the constant region of the heavy chain of the antibody. Thiolated anti-CD38 antibodies were then combined with maleimide-functionalized PEO-*b*-PBCL, such that antibodies were attached to the surface of the polymers. Lastly, anti-CD38-conjugated polymers were mixed with NP loaded with S3I-1757 (i.e., S3I-NP) to generate CD38-S3I-NP.

We then determined if the conjugation of anti-CD38 on the surface of S3I-NP significantly altered their physical properties. As summarized in Table 1, the average size of CD38-S3I-NP was 91.4 ± 9.4 nm, which is not significantly different from that of S3I-NP (97.4 ± 5.2 nm) (*p* = 0.39). Similarly, there is no significant difference in drug encapsulation efficiency between CD38-S3I-NP and S3I-NP (81.6 ± 7.2% versus 87.0 ± 9.2%, *p* = 0.47) as well as drug loading (14.7 ± 1.3% versus 15.7 ± 1.7%, *p* = 0.47). The polydispersity index was significantly higher in CD38-S3I-NP compared to that of S3I-NP (0.367 ± 0.016 versus 0.273 ± 0.003, *p* < 0.001), suggesting that CD38-S3I-NP is less uniform in size compared to S3I-NP, possibly due to antibody aggregation. As shown in Figure 1B, significantly more S3I-1757 was found to be released from CD38-S3I-NP than that from S3I-NP after 1, 2, and 4 hours of incubation (*p* < 0.001, *p* < 0.001, and *p* = 0.002, respectively). Nevertheless, both formulations reached a comparable amount of S3I-1757 release (~68%, *p* = 0.59) at 24 h. Taken together, the physical properties between these two formulations are not substantially different.

### 2.2. Anti-CD38 Conjugation on NP Results in More Cellular Uptake by MM Cells

We then determined if the conjugation of anti-CD38 to NP can significantly improve the uptake of NP by MM cells. To facilitate the detection and quantification of NP in vitro, we synthesized Cy5.5 (a fluorophore)-conjugated NP with or without the coating of anti-CD38 (denoted as Cy5.5-CD38-NP and Cy5.5-NP, respectively). The NP used in these experiments was not loaded with the STAT3 inhibitor to avoid drug-induced cytotoxicity, which can potentially interfere with our assays. Two MM cell lines (U266 and RPMI8226) were used. SupM2, an ALK-positive anaplastic large cell lymphoma cell line, was used as a negative control. The CD38 expression in the two MM cell lines and the absence of CD38 expression in SupM2 are illustrated in Appendix A. As shown in Figure 2, both MM cell lines incubated with Cy5.5-CD38-NP for 4 h exhibited a significantly higher level of intracellular Cy5.5 compared to cells incubated with Cy5.5-NP. Specifically, in U266 cells, Cy5.5-CD38-NP treatment yielded 43.2 ± 0.1% Cy5.5-positive cells, whereas Cy5.5-NP treatment resulted in only 0.4 ± 0.1% Cy5.5-positive cells (*p* < 0.001). Similarly, in RMMI8226 cells, Cy5.5-CD38-NP yielded significantly more Cy5.5-positive cells than Cy5.5-NP treatment (76.7 ± 1.1% versus 1.2 ± 0.1%) (*p* < 0.001). Compared to the background (i.e., no treatment), Cy5.5-CD38-NP only minimally increased the proportion of Cy5.5-positive cells in SupM2 cells (9.2 ± 0.3%).

### 2.3. CD38-S3I-NP is More Cytotoxic to MM Cells than S3I-NP

We next assessed the cytotoxicity of CD38-S3I-NP compared to S3I-NP in two MM cell lines (U266 and RPMI8226) using MTS assay. In these experiments, we added exogenous IL6 (2 ng/mL) to the cell culture to enhance STAT3 activity in the cells. As shown in Figure 3A, in RPMI8226 cells, the addition of CD38-S3I-NP led to significantly lower cell viability compared to S3I-NP at 48 h (50 μM, *p* = 0.001). In U266 cells, significantly lower cell viability of CD38-S3I-NP was seen at 100 μM compared to S3I-NP (*p* = 0.007). In contrast, there was no significant difference in reducing cell viability of SupM2 between the two formulations, although these cells are known to be highly STAT3-active due to an endogenous tyrosine kinase, NPM-ALK [25]. As a comparison, we repeated the experiments using the combination of S3I-NP and free CD38 antibodies. As shown in Figure 3A, the results were similar to those of S3I-NP and significantly inferior to those CD38-S3I-NP in the two MM cell lines. The inhibitory concentration at 50% (IC_50_) values generated from all of these experiments is summarized in Table 2. In the same experiments, we also confirmed that CD38-S3I-NP and S3I-NP were effective in suppressing STAT3 phosphorylation at residue Y705 (i.e., pSTAT3). Using Western blot analysis, we found that pSTAT3 induced by IL6 (2 ng/mL) in the two MM cell lines was substantially decreased by both formulations at 24 h (Figure 3B).

### 2.4. CD38-S3I-NP is More Effective in Suppressing MM Tumor Growth In Vivo Compared to S3I-NP

We then elucidated if CD38-S3I-NP has therapeutic advantages over S3I-NP in a SCID mouse xenograft model. As detailed in Materials and Methods, we xenografted U266 cells stably expressing a luciferase expression construct (U266-luc) in SCID mice, such that the growth of tumors can be easily tracked ex vivo using bioluminescence imaging. When the tumors became detectable, S3I-NP or CD38-S3I-NP was injected intravenously. On day 15 after the injection of NP, 4/4 animals in the S3I-NP group reached the endpoints as defined in Materials and Methods, while 1/4 animals in the CD38-S3I-NP group did not. Statistical analysis reveals a trend for a longer survival for the CD38-S3I-NP group, although the difference between the two groups does not reach statistical significance (*p* = 0.079, Mantel-Cox test), most likely due to the small sample size. As a control, both mice treated with phosphate buffered saline (PBS) reached the endpoints on day 12.

Other than the time needed to reach the endpoints, we also directly assessed tumor growth in the two study groups. Specifically, we summed up the levels of detectable bioluminescence (radiance ranged between 4.00 × 10^5^ and 1.00 × 10^7^ p/sec/cm^2^/scr) by using the IVIS Spectrum In Vivo Imaging System, as described in Materials and Methods. Images of a representative animal from each of the CD38-S3I-NP, S3I-NP, and PBS groups are shown in Figure 4A. Animals in the CD38-S3I-NP group had significantly lower tumor volume compared to the S3I-NP group at 240 and 288 h (*p* = 0.018 and *p* = 0.006, respectively). Since we had only two animals in the PBS group, the statistical significance cannot be determined for this group. Nonetheless, it is evident that the tumors grew substantially faster than those in the CD38-S3I-NP group (tumor volume at 288 h was 16.8 times higher).

We then assessed if the differences in tumor growth and survival between the CD38-S3I-NP and S3I-NP groups correlates with a difference in STAT3 down-regulation. To achieve this goal, we extracted bone marrow cells from specific bone fragments in which the involvement by MM was confirmed by the expression of bioluminescence. The expression of pSTAT3 was then detected using Western blot analysis and immunocytochemistry. The vast majority of cells extracted from the lesions (detailed in Materials and Methods) were confirmed to MM cells morphologically by using CD38 immunocytochemistry. pSTAT3 immunocytochemistry was then performed, and we found that MM cells from the CD38-S3I-NP group had no or barely detectable pSTAT3 signals, whereas MM cells from the S3I-NP group had relatively strong pSTAT3 signals in most of the cells examined (Figure 4B). This difference in pSTAT3 expression between the two groups was further confirmed by Western blot analysis (Figure 4C).

## 3. Discussion

It has been demonstrated that a number of novel NP drug delivery systems can significantly improve drug bioavailability in experimental models [26]. The mechanism for this improvement is believed to be attributed to the large size of the NP, which can pass through the leaky tumorous vasculature but not the normal blood vessels, resulting in the preferential accumulation of these NP in the tumors [26]. Many NP also allow the encapsulation of hydrophobic drugs which otherwise cannot be delivered to the cellular targets in their free form [27]. In this regard, our group had previously developed an NP which can effectively encapsulate and deliver cucurbitacin, a naturally occurring STAT3 inhibitor, making it more compatible for clinical use [28]. More recently, we used the same NP to encapsulate S3I-1757, a newly developed STAT3 inhibitor with higher potency and specificity, and we found that this NP (i.e., S3I-NP) demonstrated significant therapeutic efficacy against melanoma in a SCID mouse xenograft model [24].

To further improve the therapeutic potential of NP in treating cancer, various researchers have attempted to decorate NP with various targeting moieties such as monoclonal antibodies. For example, in a subcutaneous breast cancer xenograft mouse model, it was found that trastuzumab (an anti-HER2 antibody)-conjugated NP carrying doxorubicin was accumulated in the xenografts better than NP without antibodies, and this finding correlated with a >50% improvement in the reduction of tumor volume [7]. Similarly, in two other studies, cetuximab (an anti-EGFR antibody)-conjugated NP loaded with paclitaxel or gemcitabine was also found to show superior efficacy and tumor-targeting effects compared to NP without antibody conjugation [29,30]. Our group has previously found that anti-CD30 conjugated to a commercially available liposomal doxorubicin (Doxil^®^) was significantly more effective in treating ALK-positive anaplastic large cell lymphoma in a SCID mouse xenograft model [31]. In addition to the targeting function, antibodies conjugated on the surface of NP are also believed to prevent the uptake/removal of the drugs by the reticuloendothelial system [32].

The testing of STAT3 inhibitors in treating MM has been previously attempted. Previous studies have reported that novel STAT3 inhibitors such as atovaquone, SC99, and LLL12 can kill STAT3-active MM cells and significantly suppress subcutaneous tumors in SCID-mouse xenograft models [33,34,35]. However, these new anti-STAT3 agents appear to be structurally hydrophobic; without a carrier, their clinical uses will be limited as they are water-insoluble. Regarding CD38, we believed that this is a good target for MM because it is highly expressed in most cases of MM and its expression is relatively restricted in normal cells [36]. CD38 has been regarded as a therapeutic target for MM, and daratumumab is the first human anti-CD38 antibody approved by the Food and Drug Administration for treating MM [37,38]. To the best of our knowledge, our study is the second to report the development of an antibody-conjugated NP to treat MM. The first study was reported in early 2018, in which anti-CD38 conjugated NP carrying bortezomib exhibited a 2 to 3-fold increase in cell uptake by MM cells and a ~50% greater reduction of MM tumor growth compared to non-targeted NP [39]. These results are in line with our findings. Taken together, we believe that NP carrying a potent STAT3 inhibitor (such as S3I-1757) decorated by anti-CD38 is a reasonable approach to treat MM. Our data is in support of this concept.

The method we employed to conjugate anti-CD38 onto NP involved thiolation of the lysine residues of the constant region of immunoglobulins. The thiolated antibody then formed a highly stable thioester bond with the maleimide polymers. We believe that our conjugation method can provide two main advantages compared to that used in the other study of using anti-CD38 conjugated NP to treat MM, in which anti-CD38 was attached to NP via biotinylation [39]. First, the covalent thioester bond between the antibody and NP is substantially more stable than the non-covalent biotinylation bond, which likely results in a longer half-life of NP in vivo. Second, compared to biotinylation, the thiolation process of anti-CD38 is limited to its constant region, thus minimizing the risk of re-directing the antibodies in the wrong orientation (i.e., the hypervariable region of the immunoglobulin pointing inward). Thus, we believe that thiolation of the immunoglobulin can confer better stability and therapeutic efficacy to the NP.

Due to the NP barrier and the superior MM cell-targeting ability, we speculated that CD38-S3I-NP would possess a higher maximal tolerable dose and lower incidences of adverse effects compared to S3I-NP and free S3I-1757. Unlike OPB-31121, CD38-S3I-NP is not orally available since it can dissemble under conditions of adverse pH within the gastrointestinal tract. Thus, intravenous injection will be the best method to administer CD38-S3I-NP. Because STAT3 activity is linked to resistance to chemotherapy in MM, combined therapy of bortezomib, thalidomide, or dexamethasone with CD38-S3I-NP may improve response rates. One possible caveat of using CD38-S3I-NP in clinical settings is the fast clearance through the liver and kidneys. Therefore, approaches which prevent rapid NP clearance have to be developed to avoid toxicity to these organs.

## 4. Materials and Methods

### 4.1. Materials and Cell Culture

S3I-1757 (white powder with a molecular weight of 521.6 g/mol, soluble in DMSO) was obtained from Glixx Laboratories (Hopkinton, MA, USA). Methoxy-PEO (average molecular weight of 5000 g/mol), diisopropylamine (99%), benzyl chloroformate (tech 95%), sodium (in kerosin), butyllithium in hexane (2.5 M solution), palladium-coated charcoal, pyrene, and Cremophor^®^ EL were purchased from Sigma Aldrich (St. Louis, MO, USA). α-benzyl carboxylate ε-caprolactone was prepared by Alberta Research Chemicals Incorporation (Edmonton, AB, Canada). Stannous octoate was purchased from MP Biomedicals Incorporation (Solon, OH, USA). All other chemicals were reagent grade. U266 and SupM2 cells were purchased from American Type Culture Collection (Manassas, VA, USA). RPMI8226 cells are a gift from Dr. Linda Pilarski (Department of Oncology, University of Alberta, Edmonton, AB, Canada). The cells were cultured in RPMI 1460 medium with L-glutamine (Life Technology, Carlsbad, CA, USA), 10% FBS (Life technology) and, 100U/mL penicillin/streptomycin (Sigma Aldrich). All cells were incubated at 37 °C supplied with 5% atmospheric CO_2_.

### 4.2. Purification of Anti-CD38

The hybridoma cells (TBH-7) producing humanized anti-CD38 were cultured in RPMI1460 medium supplied with 10% fetal bovine serum. When the cells were confluent, they were transferred to RPMI1460 medium with 10% ultra-low IgG fetal bovine serum (ThermoFisher, Waltham, MA, USA). The cell supernatant was collected after 48 h. 150 mL of supernatant was concentrated to 5 mL using Amicon Ultra-15 Centrifugal Filter Unit (Millipore, Burlington, MA, USA). The concentrated supernatant was incubated in NAb™ Protein A/G Spin Column (ThermoFisher) for 10 min. The bound antibody was eluted out using the elution buffer (ThermoFisher). The concentration of purified anti-CD38 was measured by NanoDrop™ 1000 Spectrophotometer (ThermoFisher). The purified anti-CD38 was dialyzed in sterile PBS overnight prior to NP conjugation.

### 4.3. Preparation of NP

Poly-(ethylene oxide)-block-poly-(α-benzyl carboxylate ε-caprolactone) (PEO-b-PBCL) was synthesized using the method previously described [40]. In brief, α-benzyl carboxylate ε-caprolactone was mixed with methoxy-poly-(ethylene oxide) at 1:1.12 weight ratio with a trace amount of stannous octoate. The reaction mixture was incubated for 4 h at 140 °C in the vacuum oven and stopped by cooling the reaction at room temperature overnight. S3I-NP was prepared from PEO-b-PBCL block copolymers as previously described [24]. In brief, 20 mg of block copolymer was dissolved in tetrahydrofuran and was mixed with 2 mg S3I-1757 dissolved in DMSO. The mixture was incubated at room temperature with stirring overnight. The excess S3I-1757 was removed by centrifugation. For anti-CD38 conjugation, anti-CD38 was mixed with 2-imidothiolane at room temperature at pH 8.0 to synthesize thiolated anti-CD38. The maleimide PEO-b-PBCL was prepared by following a previously established protocol [41]. Micellized maleimide PEO-b-PBCL was mixed with thiolated anti-CD38. The anti-CD38 conjugated NP were mixed with S3I-NP in water to form CD38-S3I-NP through post-insertion method. The size and polydispersity index of S3I-NP and CD38-S3I-NP were measured by Zetasizer Nano^®^. The S3I-1757 concentrations in CD38-S3I-NP and S3I-NP were measured by high-performance liquid chromatography (HPLC) using a previously established protocol [24]. For the synthesis of Cy5.5-conjugated NP, a previously described protocol was followed [42].

### 4.4. In Vitro Release Assay

In vitro release assay was carried out as previously described [24]. In brief, 1 mL of CD38-S3I-NP or S3I-NP was put in a semi-permeable dialysis bag (molecular weight cutoff: 12,000–14,000 kDa). The bag was placed in sterile PBS and incubated in a shaking water bath at 37 °C. A 50 µL aliquot from the dialysis bag was collected at various time points for S3I-1757 concentration measurement by HPLC analysis. To maintain the total volume, 50 µL of PBS was added back to the dialysis bag after aliquot collection.

### 4.5. Cellular Uptake Assay

NP chemically conjugated with Cy5.5 (an amount equivalent to 0.2 ng Cy5.5) was added to 1.0 × 10^6^ U266, RPMI8226 and SupM2 cells and cultured at 37 °C in dark for 4 h. Cells were washed with sterile PBS twice and subjected to flow cytometry (BD FACSCantoII, Franklin Lakes, NJ, USA) analysis using the Per-Cy5.5-APC fluorescence channel.

### 4.6. Cell Viability Assay

Cell proliferation was assessed by the CellTiter 96^®^ AQueous one solution cell proliferation MTS assay (Promega, Madison, WI, USA). Approximately 2.5 × 10^4^ U266 or RPMI8226 cells were seeded in each well of a 96-well plate and treated for 24 or 48 h, and 20 µL 3-(4,5-dimethylthiazol-2-yl)-5-(3-carboxymethoxyphenyl)-2-(4-sulfophenyl)-2H-tetrazolium (MTS) was added to each well. The absorbance of light at 490 nm was measured by a FLUOstar OPTIMA microplate reader (BMG Labtech, Cary, NC, USA). The IC_50_ values were calculated by Graph Prism 7 from the cell viability versus the logarithm of the concentration curve.

### 4.7. Western Blot Analysis

Total cell lysates were prepared and lysed with 1× RIPA buffer (10× stock solution from Cell Signaling Technology, Danvers, MA, USA) with 0.05% protease inhibitor cocktail (Millipore) and 0.05% phosphatase inhibitor cocktail (Millipore). Protein concentrations were measured using a Pierce™ BCA Protein Assay Kit (ThermoFisher). Cell lysates treated with SDS were subject to SDS-PAGE and transferred onto a nitrocellulose membrane. The membrane was probed with anti-STAT3 (1:1000, CST, #9139), anti-pSTAT3 (Y705) (1:2000, CST, #9145), and anti-β-actin (1:1000, CST, #58169) diluted in 5% BSA in TBS-Tween20 (0.05%, *v*/*v*). These antibodies were probed with anti-mouse IgG conjugated with horseradish peroxidase (1:1000, Cell Signaling). The membrane was washed three times with TBS-T after secondary antibody treatment. The bands on the membrane were visualized with Pierce™ ECL Western blotting substrate (Thermo Scientific) and exposed to X-ray films (Fuji, Tokyo, Japan).

### 4.8. Preparation of U266-luc Cells by Lentiviral Transduction

Lentiviral particles carrying pLenti-Puro-luc were a kind gift from Dr. Kyle Potts and Dr. Mary Hitt (Department of Oncology, University of Alberta, Edmonton, AB, Canada). Also, 2 × 10^6^ U266 cells were transduced with 2 mL of lentiviral particles and 6 µg/mL polybrene on a 6-well plate after 30 min of spin inoculation at 1000× *g*. After 24 h, cells were washed and replenished with fresh lentiviral particles for another 24 h. Transduced cells were washed with PBS and resuspended in fresh growth medium for 2 days. The transduced cells were selected with 2 µg puromycin (ThermoFisher) in the medium. The luciferase activity of U266-luc cells was measured by bioluminescence imaging (Appendix A). The growth rate and responsiveness to S3I-1757 of U266-luc were confirmed to be insignificantly different from parental U266 cells (Appendix A).

### 4.9. In Vivo Studies Using MM Xenograft

The experimental protocols for all in vivo studies in this manuscript were approved by Animal Care and Use Committees, University of Alberta (#AUP00000282). Half of a million U266-luc was injected into SCID mice (Jackson, strain NOD.Cg-Prkdcscid Il2rgtm1Wjl/SzJ) intravenously. Twelve days after injection, 100 µL of S3I-NP or CD38-S3I-NP colloidal dispersions in dextrose 5 % was injected into the mice intravenously via the tail vein for two consecutive days at a dose of 3 mg/kg per day. The tumor size was measured by quantifying the total flux of bioluminescence signals detectable (radiance ranged between 4.00 × 10^5^ and 1.00 × 10^7^ p/sec/cm^2^/scr) on the ventral side of each animal at various time points using Living Image Software (PerkinElmer, Waltham, MA, USA). At the endpoint defined by the approved protocol, the animals were euthanized and the bone marrow cells with MM bone lesion (visualized by bioluminescence) were removed from the femoral bone and split into two portions. One portion was stored at −80 °C as cell pellets for Western blot analysis, and another was stored in 4% paraformaldehyde at 4 °C for immunocytochemistry.

### 4.10. Immunocytochemistry

Mouse bone marrow cells in 4% paraformaldehyde (Sigma Aldrich) were pelleted and resuspended in liquid histogel (Thermo Scientific) and transferred to a 15 × 15 mm^2^ plastic mold (Leica, Wetzlar, Germany). Upon solidification, the cell-histogel was subjected to processing and embedding. The cell blocks were sectioned into 5-μm slides. Slides were rehydrated in xylene and decreasing concentrations of ethanol. The antigens were retrieved using 1× citrate buffer (Sigma) by microwaving in a pressure cooker for 20 min. pSTAT3 (1:50, Santa Cruz, #sc-8059) and CD38 antibodies (1:100, Abcam, #ab108403) were diluted at 1:100 in antibody diluent (DAKO, Glostrup, Denmark). MACH2 mouse HRP polymer (Biocare Medical, Pacheco, CA, USA) was used as a secondary antibody. The chromogen and substrate were mixed and applied to each slide for 2 min for color development (DAKO).

### 4.11. Statistical Analysis

All numerical data in this study was presented as the mean from experiment replicates or independent experiments as described in the figure legends. For the comparison of IC_50_ values in Table 2, two-way ANOVA and Tukey’s multiple comparison test were employed. For survival analysis in the animal study, Statistical significance between groups was analyzed using the Mantel-Cox test (α = 0.05). For the remaining comparisons, a Student’s *t*-test with α = 0.05 was used for the statistical analysis. The analysis was done using Microsoft Excel 365, except for Table 2, Figure 4A, and the overall survival analysis in the animal study, for which GraphPad Prism 7 was used for analysis.

## 5. Conclusions

In summary, this study has provided the proof-of-principle that the decoration of anti-CD38 on NP loaded with STAT3 inhibitors can further improve their therapeutic effects against MM. We also predict that the use of these NP can significantly lower the unwanted side effects of STAT3 inhibitors, as they are targeted to cancer cells, and preferentially released inside of these cells. These encouraging results provide sufficient basis for a Phase I clinical trial.

## Figures and Tables

**Figure 1 cancers-11-00248-f001:**
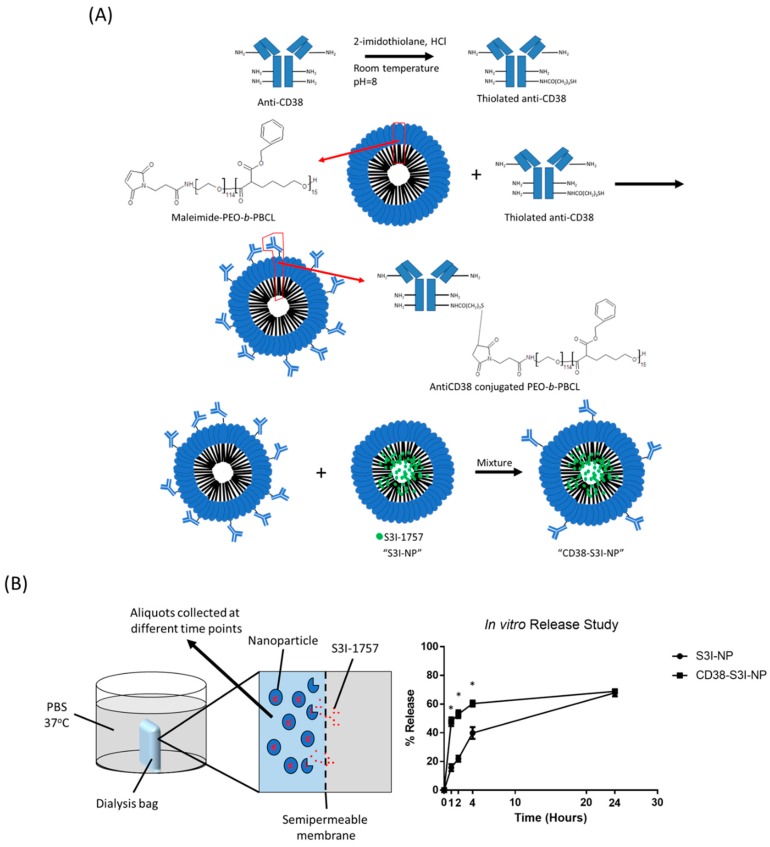
The synthesis of CD38-S3I-NP. (**A**) Chemical reactions of the anti-CD38 conjugation to PEO-*b*-PBCL, the building block of our nanoparticles (NP). The final product was mixed with S3I-NP to generate CD38-S3I-NP. (**B**) The release of S3I-1757 from S3I-NP or CD38-S3I-NP in vitro. The percentage of S3I-1757 released was calculated by the lost amount of S3I-1757 compared to the initial total amount of S3I-1757. The error bar represents the standard deviation from a triplicate experiment; * *p* < 0.05, via Student’s *t*-test.

**Figure 2 cancers-11-00248-f002:**
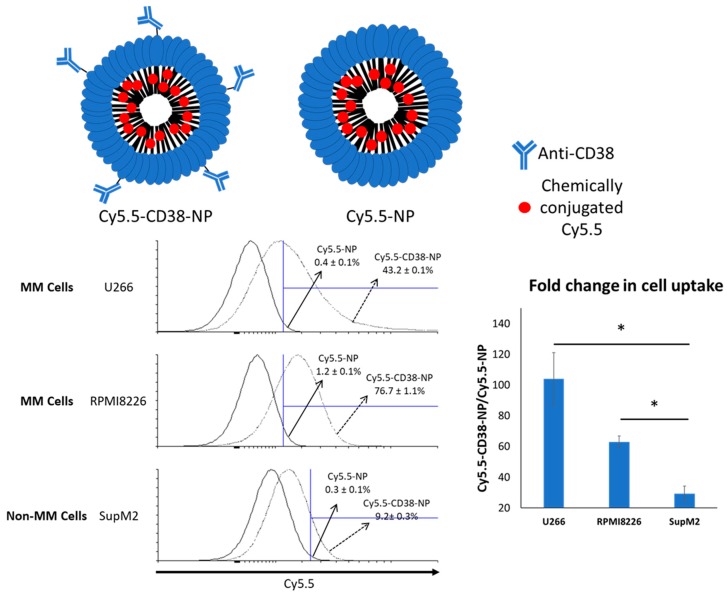
Flow cytometry analysis of the Cy5.5-positive cell population 4 h after treatment of Cy5.5-NP or Cy5.5-CD38-NP. Anti-CD38-conjugated NP exhibits improved cellular uptake of NP by multiple myeloma (MM) cells. Cy5.5 was chemically conjugated to the core of NP. The gated area was defined using the cells without NP treatment. The representative dot plot from a triplicate experiment is shown. The error values represent the standard deviation from the triplicate experiment. A non-MM cell line, SupM2, was included for comparison. The fold change in cell uptake was calculated by dividing the percentage of Cy5.5-positive cells with Cy5.5-CD38-NP treatment by that with Cy5.5-NP treatment. The error bar represents standard deviation from a triplicate experiment; * *p* < 0.05, via Student’s *t*-test.

**Figure 3 cancers-11-00248-f003:**
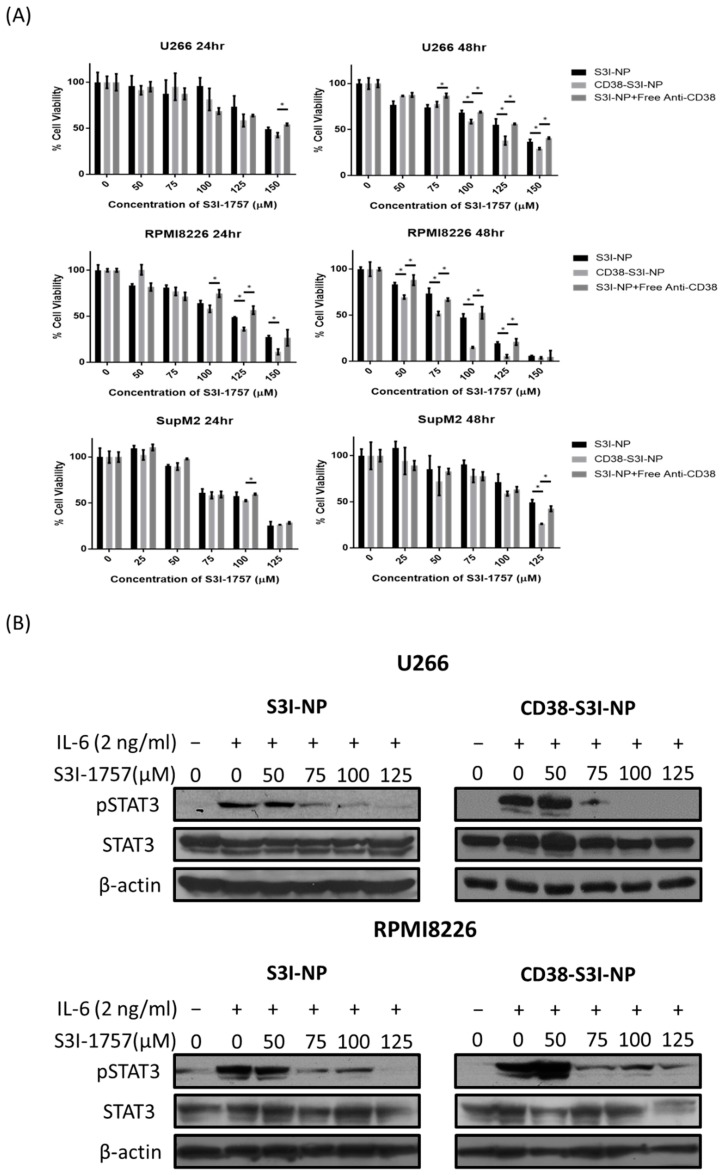
CD38-S3I-NP induces cytotoxicity and inhibits STAT3 activity in MM cells. (**A**) U266 and RPMI8226 cells were then treated with S3I-NP, CD38-S3I-NP, or S3I-NP with free CD38 antibody at a concentration which is equivalent to CD38-S3I-NP (1.4 mg/mL) with the presence of IL6 (2 ng/mL) for 24 and 48 h. Cell viability was measured using a 3-(4,5-dimethylthiazol-2-yl)-5-(3-carboxymethoxyphenyl)-2-(4-sulfophenyl)-2H-tetrazolium (MTS) cell viability assay in triplicate; * *p* < 0.05, via Student’s *t*-test. (**B**) Western blot analysis of STAT3 and pSTAT3 levels in U266 and RPMI8226 cells treated with S3I-NP or CD38-S3I-NP with the presence of IL6 (2 ng/mL) for 24 h. β-actin was blotted as a loading control.

**Figure 4 cancers-11-00248-f004:**
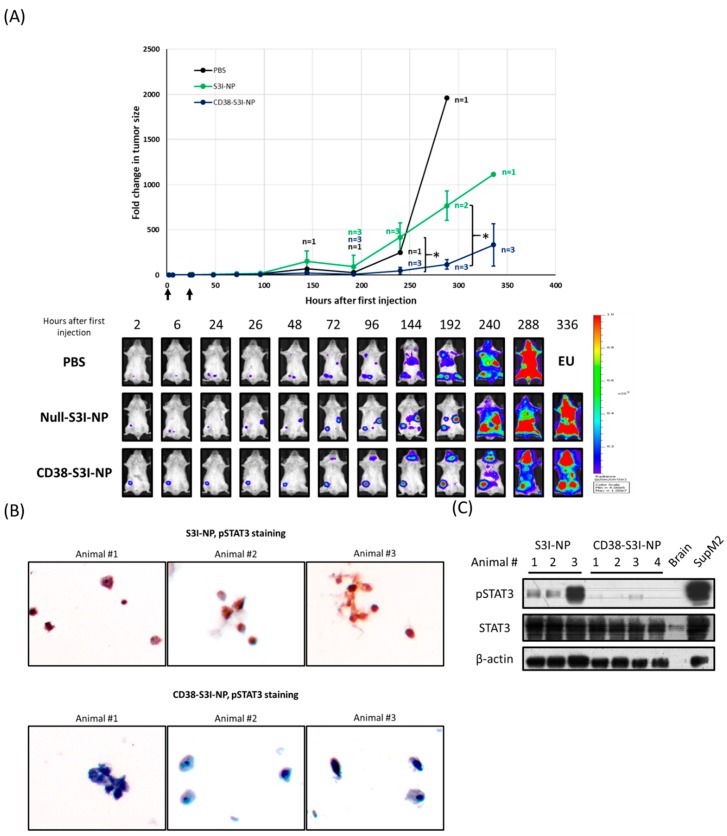
CD38-S3I-NP is more tumor suppressive than S3I-NP in MM xenograft. (**A**) SCID mice intravenously injected with PBS (black line, *n* = 2), 3 mg/kg S3I-NP (green line, *n* = 4) or CD38-S3I-NP (blue line, *n* = 4) every day for two days as indicated by arrows on the *x*-axis. Animal numbers other than the initial numbers at different time points are indicated. The tumor size was quantified by the bioluminescence intensity and normalized to the initial bioluminescence signal (i.e., 2 h post-injection). The representative bioluminescence images of animals treated with PBS, S3I-NP, or CD38-S3I-NP were shown. * *p* < 0.05, via Student’s *t*-test; EU—euthanized. (**B**) The pSTAT3 levels in bone marrow mononuclear cells extracted from the SCID mice in (**A**) at the endpoint. Non-tumorous brain tissue from a SCID mouse was used as a negative control. SupM2 cells were used as a positive control for pSTAT3. β-actin was blotted as a loading control. (**C**) Immunocytochemical staining of pSTAT3 and in bone marrow mononuclear cells from (**B**). Each image represents bone marrow cells from one animal. The images were taken at a magnification of 400×.

**Table 1 cancers-11-00248-t001:** Physical properties of S3I-NP and CD38-S3I-NP.

NP Formulation	Average Size (nm)	Polydispersity Index	Drug Encapsulation Efficiency (%)	Drug Loading (Weight %)
S3I-NP	97.4 ± 5.2	0.273 ± 0.003	87.0 ± 9.2%	15.7 ± 1.7%
CD38-S3I-NP	91.4 ± 9.4	0.367 ± 0.016*	81.6 ± 7.2%	14.7 ± 1.3%

* *p* < 0.05, compared to S3I-NP.

**Table 2 cancers-11-00248-t002:** IC_50_ values of U266, RPMI8226, and SupM2 cells treated with different NP.

Cell Line	Treatment	IC_50_ (μM)	
		24 h	48 h
U266	S3I-NP	136.7–163.6	115.4–148.6 **
	CD38-S3I-NP	127.7–151.3	106.3–114.0
	S3I-NP + Free Anti-CD38	143.5–172.4	128.6–139.4 **
RPMI8226	S3I-NP	110.9–124.5	88.2–98.1 **
	CD38-S3I-NP	100.4–109.8	64.0–73.6
	S3I-NP + Free Anti-CD38	108.9–142.2	87.2–99.5 **
SupM2	S3I-NP	88.8–105.6	110.4–144.8
	CD38-S3I-NP	86.0–98.9	83.16–119.4
	S3I-NP + Free Anti-CD38	90.1–110.0	106.9–134.9

** *p* < 0.0001, compared to the corresponding CD38-S3I-NP treatment.

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
