# Peer review of "Decoration of Anti-CD38 on Nanoparticles Carrying a STAT3 Inhibitor Can Improve the Therapeutic Efficacy Against Myeloma"

_cancers, 2019, doi:10.3390/cancers11020248_

Reviewer 1 Report

This is an exciting and innovative report. The authors are to be congratulated. Minor issues:

- Can the authors speculate on the tolerance of the complex ina human therapeutic setting?

- Which specific toxicities are to be expected?

Author Response

Comments and Suggestions for Authors

This is an exciting and innovative report. The authors are to be congratulated. Minor issues:

1)   Can the authors speculate on the tolerance of the complex in a human therapeutic setting?

REPLY: We appreciate the reviewer’s compliment. As suggested, we have added the discussion on CD38-S3I-NP in the aspects of therapeutic setting regarding tolerability, method of administration, drug combination and caveats/toxicity in the last paragraph of Discussion (page 9 and 10).

2)   Which specific toxicities are to be expected?

REPLY: Please see above.

Reviewer 2 Report

In this manuscript, Huang et al. generatedCD-38-S3I-NP and showed that it was efficient in suppressing STAT3 phosphorylation in vitro and in vivo. This is a good study. I have only several minor issues.

1.    In line 89 the p value is 0.39 but in table 1 the p value is<0.05. And I am curious why addition of CD38 antibody makes the size of the nanoparticles smaller.  

2.    In Figure 3A, 48-hour experiments are shown but in the legend it says only 24 hours.

3.    MTS assay is more likely a proliferation assay and it is not very appropriate to be used to determine viability.

4.    In Figure 3B, the addition CD38-S3I-NP further increases STAT3 phosphorylation. And it seems like CD38-S3I-NP is less efficient than S3I-NP in inhibiting STAT3 phosphorylation.

Author Response

Comments and Suggestions for Authors

In this manuscript, Huang et al. generatedCD-38-S3I-NP and showed that it was efficient in suppressing STAT3 phosphorylation in vitro and in vivo. This is a good study. I have only several minor issues.

1)   In line 89 the p value is 0.39 but in table 1 the p value is<0.05. And I am curious why addition of CD38 antibody makes the size of the nanoparticles smaller. 

REPLY: We thank the reviewer for noticing this. The sizes of S3I-NP and CD38-S3I-NP is NOT significantly different from each other. We have removed the asterisk mark in Table 1 in the Size column (page 3). We believed that the slightly lower size of CD38-S3I-NP is likely due to loss of stability during the process of insertion of anti-CD38 conjugated polymers.

2)   In Figure 3A, 48-hour experiments are shown but in the legend it says only 24 hours.

REPLY: We thank the reviewer for pointing this out. We have amended the figure legends of Figure 3A (“24 hours” to “24 and 48 hours”) (page 6).

3)   MTS assay is more likely a proliferation assay and it is not very appropriate to be used to determine viability.

REPLY: According to the manufacturer (Promega), the MTS assay we used can be used to determine the number of viable cells which is actively proliferating. It is a commonly used method for measuring the cytotoxicity of compounds to cells.

4)   In Figure 3B, the addition CD38-S3I-NP further increases STAT3 phosphorylation. And it seems like CD38-S3I-NP is less efficient than S3I-NP in inhibiting STAT3 phosphorylation.

REPLY: We quantified the band intensity of STAT3 and pSTAT3, calculated the ratio of pSTAT3/STAT3, and normalized it to IL6 only treatment. We found that the treatment of both S3I-NP or CD38-S3I-NP did not increase STAT3 phosphorylation compared to the controls. In terms of STAT3-inhibiting efficiency, we believed that it is hard to conclude which formulation is more effective based on one western blot experiment because S3I-NP and CD38-S3I-NP have different pharmacokinetics (i.e. drug release rates, cellular uptake rates, etc). Our main purpose of Figure 3B was to demonstrate that CD38-S3I-NP is capable of inhibiting STAT3 phosphorylation.

Reviewer 3 Report

This excellent work shows that STAT3 inhibitor nanoparticles directed with CD38 antibodies against MM cells have a superior efficacy in preclinical (in vitro as well as in vivo) models of MM. The results are highly convincing and the conclusions drawn are comprehensible. The techniques used are sophisticated. The linguistic style is very good making the manuscript easy to read and understand. This manuscript is a worthy addition to the scientific literature and will find many interested readers. Therefore I strongly advocate publication in "Cancers".

To my estimation this work is a sufficient basis for a phase I clinical study. Therefore I suggest to strengthen the last sentence of the "Conclusions" part of the manuscript.

Author Response

Comments and Suggestions for Authors

This excellent work shows that STAT3 inhibitor nanoparticles directed with CD38 antibodies against MM cells have a superior efficacy in preclinical (in vitro as well as in vivo) models of MM. The results are highly convincing and the conclusions drawn are comprehensible. The techniques used are sophisticated. The linguistic style is very good making the manuscript easy to read and understand. This manuscript is a worthy addition to the scientific literature and will find many interested readers. Therefore I strongly advocate publication in "Cancers".

To my estimation this work is a sufficient basis for a phase I clinical study. Therefore I suggest to strengthen the last sentence of the "Conclusions" part of the manuscript.

REPLY: The authors appreciate the positive comments from the reviewer. As suggested, we changed the last sentence of Conclusion (page 12) to make it stronger.

Round  2

Reviewer 2 Report

I have no more issues about the revised manuscript.